# Fabrication of Durable Ordered Ta_2_O_5_ Nanotube Arrays Decorated with Bi_2_S_3_ Quantum Dots

**DOI:** 10.3390/nano9101347

**Published:** 2019-09-20

**Authors:** Mateusz A. Baluk, Marek P. Kobylański, Wojciech Lisowski, Grzegorz Trykowski, Tomasz Klimczuk, Paweł Mazierski, Adriana Zaleska-Medynska

**Affiliations:** 1Department of Environmental Technology, Faculty of Chemistry, University of Gdansk, 80-308 Gdansk, Poland; mateuszadam96@gmail.com (M.A.B.); pawel.mazierski@ug.edu.pl (P.M.); 2Institute of Physical Chemistry, Polish Academy of Science, Kasprzaka 44/52, 01-224 Warsaw, Poland; wlisowski@ichf.edu.pl; 3Faculty of Chemistry, Nicolaus Copernicus University, 87-100 Torun, Poland; tryki@umk.pl; 4Department of Solid State Physics, Faculty of Applied Physics and Mathematics, Gdansk University of Technology, 80-233 Gdansk, Poland; tomasz.klimczuk@pg.edu.pl

**Keywords:** heterogeneous photocatalysis, ordered Ta_2_O_5_ nanotubes, Bi_2_S_3_ quantum dots

## Abstract

One of the most important challenges in the fabrication of ordered tantalum pentaoxide (Ta_2_O_5_) nanotube arrays (NTs) via the electrochemical method is the formation of nanotubes that adhere well to the Ta substrate. In this paper, we propose a new protocol that allows tight-fitting Ta_2_O_5_ nanotubes to be obtained through the anodic oxidation of tantalum foil. Moreover, to enhance their activity in the photocatalytic reaction, in this study, they have been decorated by nontoxic bismuth sulfide (Bi_2_S_3_) quantum dots (QDs) via a simple successive ionic layer adsorption and reaction (SILAR) method. Transmission electron microscopy (TEM) analysis revealed that quantum dots with a size in the range of 6–11 nm were located both inside and on the external surfaces of the Ta_2_O_5_ NTs. The effect of the anodization time and annealing conditions, as well as the effect of cycle numbers in the SILAR method, on the surface properties and photoactivity of Ta_2_O_5_ nanotubes and Bi_2_S_3_/Ta_2_O_5_ composites have been investigated. The Ta_2_O_5_ nanotubes decorated with Bi_2_S_3_ QDs exhibit high photocatalytic activity in the toluene degradation reaction, i.e., 99% of toluene (C_0_ = 200 ppm) was degraded after 5 min of UV-Vis irradiation. Therefore, the proposed anodic oxidation of tantalum (Ta) foil followed by SILAR decorating allows a photocatalytic surface, ready to use for pollutant degradation in the gas phase, to be obtained.

## 1. Introduction

An increased level of air pollutants requires new materials and methods to be developed, which can be applied for gas phase treatment. Heterogeneous photocatalysis seems to be an example of green technology that is still being developed, and current research in this field is focused on finding novel active photocatalysts [1]. Ta_2_O_5_ is a wide-band-gap (E_g_) semiconductor (E_g_ = 3.9 eV) [2], which could be an alternative for commonly used titanium dioxide. Although pristine Ta_2_O_5_ can only be excited by UV irradiation with a wavelength below 318 nm, its advantage is the high photoactivity attributed to its band structure, i.e., relatively higher conduction band position than that of TiO_2_, which can result in the generation of electrons with a higher potential [3]. Hence, Ta_2_O_5_ looks like a good candidate for the photocatalytic decomposition of pollutants in the gas and aqueous phases [4] or for hydrogen evolution [4,5]. Moreover, Ta_2_O_5_ could be applied as corrosion-resistant material [6], thin-film catalyst [7], anti-reflective coating [8,9], an element of ultra-large-scale integrated circuits [10], or semiconductor precursor [11]. Nanostructured Ta_2_O_5_, in the form of a powder or thin film, can be fabricated by the sol–gel approach [9], hydrothermal method [12], chemical vapour deposition [13], or anodic oxidation of tantalum foil [14,15]. The superiority of powdered Ta_2_O_5_ comes from its well-developed surface area, but powdered nanoparticles are difficult to separate from the reaction environment after the photocatalytic process and, consequently, it is difficult to reuse those photocatalysts. Otherwise, nanotube arrays obtained by the anodization of Ta foil could be easily removed from the reaction medium, regenerated, and re-used. Therefore, Ta_2_O_5_ NTs, obtained via the anodic oxidation of tantalum foil, are promising materials that can be used as a photocatalyst to decompose the noxious substances present in the gas phase. Furthermore, the electrochemical method of NT preparation is simple, repeatable, and low-cost [5], and it allows the nanotube dimension to be controlled by differing the process conditions. Additionally, it allows ordered NTs characterized by a highly developed surface area and high stability to be obtained during photocatalytic processes, as well as representing a possibility of regeneration after photocatalytic tests [16]. 

Nonetheless, the main disadvantage of Ta_2_O_5_ NTs is their weak adhesion to the substrate, i.e., to the tantalum foil [17,18]. This problem was already observed in the first report, which mentioned the anodization of tantalum foil [19], and has not been solved. Therefore, the synthesis of nanotubes tightly adhering to the surface is one of the main challenges in the work aimed at developing ordered nanotubes made of tantalum oxide for photocatalytic purposes. There are several factors affecting surface properties of Ta_2_O_5_ nanotubes obtained through anodic oxidation, such as the (i) electrolyte composition [20], (ii) time, temperature, and applied voltages during anodization [21], and (iii) annealing condition [5]—all of which could alter the adhesion of NTs to the Ta base. As mentioned above, the electrolyte composition plays a crucial role in the NT mechanism formation and it was found that the presence of F^−^ ions, responsible for chemical etching, is necessary for the formation of a nanotubular structure. Sulphuric acid (H_2_SO_4_), compared to other acids, is characterized by thermodynamic stability, and it is responsible for the homogeneous growth of nanotubes [22]. Sodium fluoride and fluoric acid are the most common compounds used as sources of F^−^. These anions allow [TaF_7_]^2−^ to be created during the growth of nanotubes. The electrolyte solution should also include compounds which are able to increase the viscosity of the electrolyte (e.g., ethylene glycol or glycerine) and conductivity. The synthesis parameters, such as the applied voltages, temperatures, and times of anodization, strongly influence the length of NTs and, consequently, the adhesion of nanotube arrays. A higher applied voltage during an anodic oxidation causes more intensive etching of the tantalum foil and directly influences the dissolution degree of oxide layers and elongates NTs. However, if the nanotubes are too long, they will be able to be peeled from a tantalum sheet. Therefore, it could be concluded that a suitable procedure of anodic oxidation of a tantalum sheet is key for the synthesis of an adherent layer of Ta_2_O_5_ nanotubes. Furthermore, photocatalytic activity enhancement could be reached by nanotube decoration by quantum dots, such as Bi_2_S_3_ QDs (Eg = 1.3–1.7 eV, depending on the size of QDs). This type of QD can be obtained by the SILAR method [23,24,25,26], via an aqueous colloidal route [27,28] or chemical bath deposition [29]. However, the SILAR method is the most appropriate for linking the semiconductor as nanotube arrays with QDs and allows the amount of deposited nanodots to be controlled [30]. Thus far, Bi_2_S_3_ QDs have been utilized to enhance the photoactivity of a photocatalyst in the degradation of pollutants in water [23,31] and a gaseous phase [23,32], photocatalytic hydrogen generation [31], and CO_2_ photoconversion to isopropanol [33], and as a photoelectrode to remove a polysulfide electrolyte [34].

In view of this, the effect of electrochemical oxidation parameters, such as the applied voltage value, time of anodization, and annealing conditions (atmosphere type and temperature), on the Ta_2_O_5_ nanotube morphology and adherence have been systematically investigated. To confirm the beneficial effect of nanotube modification by quantum dots, Bi_2_S_3_ QDs were deposited using between one and three SILAR cycles. To the best of our knowledge, for the first time, the Bi_2_S_3_ QDs/Ta_2_O_5_ NTs composite was obtained and tested by the decomposition of volatile organic compounds (e.g., toluene) present in the gas phase. A comprehensive analysis and correlation of synthesis parameters with surface properties of nanotubes allowed the most suitable protocol to be found for the preparation of highly adhered and highly photoactive Ta_2_O_5_ nanotubes. To evaluate the stability of the obtained composite, the efficiency of toluene degradation on the Bi_2_S_3_/Ta_2_O_5_ sample was analysed in three subsequent cycles.

## 2. Materials and Methods

Tantalum foil (0.5 mm thickness, 99.9% purity) was purchased from HMW Hauner Metallische Werkstoffe (Röttenbach, Germany) toluene, sulfuric acid 96%, glycerine 99.5%, acetone, methanol, and isopropanol were purchased P.P.H. “STANLAB” (Lublin, Poland); and ammonium fluoride 98+% was purchased from ACROS ORGANICS (Geel, Belgium).

The tantalum foil was cut into pieces of a size of 3 × 2 cm. Subsequently, pieces of Ta foil were cleaned in an ultrasonic bath in acetone, isopropanol, methanol, and deionized water (<5 µS), respectively, for 10 min in each solvent. Then, the tantalum foils were dried with an air stream. The anodising system consisted of a piece of Ta foil as a working electrode, platinum mesh as a counter electrode, and silver chloride electrode as a reference electrode. The anodization was performed in electrolytes composed of sulphuric acid (90 vol%), glycerine (5 vol%), deionized water (5 vol%), and ammonium fluoride (0.27 M) for 5 to 10 min at 10 to 20 V using a programmable DC source (MCP M10-QS1005). Then, the NTs were rinsed with deionized water, dried in air for 24 h, and cleaned in an ultrasonic bath in deionized water (0–1 min), dried in air at 80 °C for 24 h, and finally, annealed at 450–900 °C (heating rate 2 °C/min) in various atmospheres—air, NH_3,_ N_2_, or H_2_—for 1 h.

The Ta foil obtained at 15 V for 5 min was cleaned in an ultrasonic bath in deionized water for 1 h to remove the NTs layer. Then, the sheet was dried in air and anodised again in the same conditions. Prepared nanotubes were cleaned in deionized water in an ultrasonic bath (1 min), dried in air in 80 °C (24 h), and annealed at 450 °C (heating rate 2 °C/min) in N_2_ for 1 h. The Bi_2_S_3_ QDs were deposited on the NT surface via the SILAR method. The NT pieces were immersed in the 0.05 M solution of BiNO_3_ for 30 s, and then rinsed in acetone, dried, and immersed in the 0.01 M solution of sodium sulfide nonahydrate (Na_2_S∙9H_2_O). The procedure was repeated up to three times and, finally, the samples were dried at 80 °C.

The diffusion reflectance spectra (DRS) were registered with the UV-VIS Spectrophotometer (Kyoto, Japan) SHIMADZU UV-2600, in the wavelength range from 250 to 600 nm, and barium sulphate was used as a reference sample. The photoluminescence (PL) measurements were performed using a LS-50B Luminescence Spectrometer equipped with a Xenon discharge lamp. The scanning electron microscope (SEM) JEOL JSM-7610F (Tokyo, Japan), and TEM with energy-dispersive x-ray spectroscopy (EDX), FEI Europe, model TecnaiF20 X-Twin (Eindhoven, Neatherlands), were used to analyse the morphology of the obtained materials. The X-ray photoelectron spectroscopy (XPS) spectra were registered with a PHI 5000 VersaProbeTM (ULVAC-PHI) spectrometer (Chigasaki, Japan) with monochromatic Al Kλ radiation (*hν* = 1486.6 eV). The X-ray beam was focused to a diameter of 100 µm, and the measured area was defined as 250 µm × 250 µm. X-ray diffraction (XRD) measurements were used to determine the crystal structures of obtained samples. An experiment was carried out in the range of 2θ = 20°–80° with an X-ray diffractometer (Xpert PROMPD, Philips, Royston, United Kingdom) equipped with Cu Kλ radiation (λ = 1.5404 Å).

The measurement of photocatalytic activity of NTs was determined in the model reaction of the photodegradation of toluene. The experiment was carried out in a flat stainless-steel reactor (30 mL) containing a quartz window, a septum, valves, and teflon gaskets. A UV-Vis irradiation source was a Xenon lamp (Oriel 1000 W, Hamamatsu, Japan). The distance between samples and the source of light was 30 cm, and the irradiation intensity was 100 mW/cm^2^. The prepared NT sample was placed inside the reactor and toluene-polluted air (C_0_ = 200 or 400 ppm) was flowed through the reactor for 1 min. The flow rate was 30 mL/min. Then, the reactor was kept in the dark for 30 min to achieve an adsorption–desorption equilibrium. In total, 200 µL of samples was collected at 5 min intervals from the start of exposure. Analysis of the toluene concentration was performed using a gas chromatograph with a flame ionization detector and capillary column (from Restek, 30 m, with 5% diphenyl and 95% dimethyl polysiloxane). Nitrogen was used as a carrier gas (flow rate of nitrogen was 40 mL/min). The selected samples were studied in terms of toluene photodegradation under monochromatic light. Samples with a dimension of 1 × 0.5 cm were placed in the Teflon reactor with a quartz window (0.5 × 1.5 cm), filled with a gas phase containing toluene (C_0_ = 200 ppm), followed by a 30 min dark experiment (to achieve adsorption equilibrium) and 30 min irradiation by wavelengths equal to 318 nm (1.2 mW/cm^2^) and 730 nm (2 mW/cm^2^). The 1000 W Xe lamp (Oriel) equipped with a monochromator (MSH 300 LOT Quantum Design, Darmstadt, Germany) was used as an irradiation source. 

The measurements of the stability of photocatalysts were determined for the most active samples: NTs_15 V_5 min_N_2__450 °C_1 h and NTs_15 V_5 min_N_2__450 °C_1 h_QDs_SILAR_1x. The experiment was prepared as described above; however, a sample was collected at every 1.5 min interval from the start of exposure to UV-Vis irradiation for 15 min. Three processes were carried out without regeneration of the photocatalyst and three processes with the regeneration of NTs after each photocatalytic experiment.

## 3. Results 

A series of Ta_2_O_5_ NTs samples were obtained via the anodic oxidation of tantalum foil in H_2_SO_4_ solution containing F^−^ anions. The SEM images presented in Figure 1 show the effect of the applied voltage, annealing conditions (atmosphere, temperature, and time), removal of the initial layer by an ultrasonic bath, and time of anodization on the morphology and adhesive properties of the as-prepared nanotubes. Sample labelling, preparation conditions, and the dimensions of all the obtained nanotubes are presented in Table 1. The results clearly show that the external and internal diameters of the nanotubes, as well as the walls’ thickness, were similar, and they did not depend on the annealing condition. On the other hand, the length of NTs increased as the time of anodization and value of applied voltage increased. The longest nanotubes were formed when the Ta foil was anodized under 20 V for 10 min, followed by 1 h calcination at 450 °C (NTs_20 V_10 min_no_cleaned_Air_450 °C_1 h). However, these nanotubes were characterized by weak adhesion and consequently, they were easily detached from the surface, which means that they are not suitable for photocatalytic treatment of the gas phase (Figure 1a). Importantly, it was observed that the annealing temperature increased from 450 to 600 °C and then, to 750 °C, causing defragmentation of the NT arrays and their base surface tantalum foil (Figure 1b). Further tests showed that the durability of Ta_2_O_5_ nanotubes is also strongly affected by the duration of anodic oxidation. Nanotubes that formed after the application of 15 V for 5 min were uniform and strongly adhered to tantalum foil, while increasing the oxidation time to 10 min resulted in the formation of nanotubes easily disassembled from the surface (Figure 1c). The SEM images presented in Figure 1d–e confirm that time and atmosphere during annealing at 450 °C do not affect the morphology and adhesion of the NTs. Therefore, subsequent experiments were carried out for time of anodization and applied voltage, which were reduced to 5 min and 15 V, respectively, and the annealing temperature was no higher than 450 °C. Among all tested conditions, only this synthesis mode allowed a strongly adhered NT layer to be obtained, which could be easily operated in a photocatalytic reaction and regenerated. The thickness of NT arrays, which was obtained under 15 V for 5 min, is in the range of 1.68–2.19 µm. The length of nanotubes fabricated in two-step anodization is equal to 1.15 µm. 

During the anodic oxidation of tantalum foil, the current density curve was registered for samples fabricated under 10, 15, and 20 V (Appendix A). To deeply understand the formation mechanism of the NTs, SEM images of the surface were produced for the sheet, for which anodization was stopped at characteristic points of the current density curve during anodic oxidation under 15 V (Appendix A). Initially, the current density sharply declined, which is characteristic of the first stage of anodization, when the compact layer of tantalum oxide is formed. Subsequently, the second stage occurred after 20 s of anodization, when the current density increased due to reaction of the fluoric anions with tantalum oxide. After this step, a soluble complex [TaF_6_]^−^ was created and pits of nanotubes were formed. The third stage, represented as a plateau, represents the growth of nanotubes. The UV-Vis reflectance spectra registered in the range of 250–600 nm for NTs are shown in Appendix A. The nanotubes exhibited a characteristic signal in the region from 250 to 300 nm [35], which is attributed to the excitation of electrons from the valence band (VB) band to conduction band (CB), in addition to the wide peak in the region of 300–600 nm. Appendix A presents the PL spectra in the range of 300–700 nm for NTs. This experiment was carried out to estimate the electron–hole recombination rate. The more intense signal represents faster recombination of photogenerated charge carriers and consequently, it could indicate lower photocatalytic activity [36]. The signal at about 430 nm was attributed to the recombination of electrons and holes present in the traps and at the valence band edge, respectively [4]. Based on the results, it could be concluded that samples obtained under 15 V and calcined at 450 °C exhibited the lowest recombination rate. This suggested that this condition of synthesis of NTs is the most suitable for the fabrication of the most photoactive NT layer.

The elemental composition and chemical nature of elements detected in the surface layer of Ta_2_O_5_ NTs material were carefully investigated using XPS. Selected samples annealed in various gases (N_2_, NH_3_, O_2_, and air) and samples modified by Bi_2_S_3_ quantum dots (QDs) were studied. The results are summarized in Appendix A and Table 2. Only small chemical modifications in the surface layer of Ta_2_O_5_ NTs were observed after annealing in various gas atmospheres (Appendix A). A slightly larger than average surface amount of nitrogen was detected for samples annealed in N_2_ and NH_3_ (see samples NTs_15 V_5 min_N_2__450 °C_1 h and NTs_15 V_5 min_NH_3__450 °C_1 h, respectively, in Appendix A). The sample NTs_15 V_5 min_Air_450 °C_1 h, on the other hand, exhibits a larger concentration of fluorine compounds, which originated from the preparation procedure. The Ta_2_O_5_ NTs are well-characterized by the main Ti^5+^ fraction of the Ta 4f spectra (Ta 4f_7/2_ signal at 26.4 eV) [37,38,39,40]. We also note a small contribution of Ta^1+^ and Ta^0^ surface species [39,40] (Appendix A). The effect of the modification of Ta_2_O_5_ NTs by Bi_2_S_3_ QDs in the following SILAR cycles is reflected by the high-resolution (HR) spectra of Bi 4f, O 1s, and Ta 4f presented in Figure 2 and summarized in Table 2. The Bi 4f spectra revealed three chemical states of Bi Bi^0^, Bi_2_ S_3_, and Bi_2_O_3_, represented by the Bi 4f_7/2_ signals at 157.0, 158.0, and 159.5 eV, respectively [37]. Unfortunately, the Bi 4f spectra are partially overlapped by the S 2p signals attributed to sulfide and sulfate species (S 2p_3/2_ signals located at 163.5 and 168.5 eV, respectively [38]). Additionally, in the binding energy (BE) range of Ta 4f spectra, additional signals of the Bi 3d and O 2s are detected (Figure 2). However, after deconvolution, both spectra evidenced the heavy coating of the Ta_2_O_5_ by the Bi_2_S_3_ after the third SILAR cycle. The Bi 4f spectrum shows the Bi_2_S_3_ fraction as a major contribution of Bi species and the Ta 4f spectrum is overlapped by the strong Bi 5d signals. Moreover, the sulfate fraction of S 2p is not detected and instead of the main O 1s signal related to Ta_2_O_5_ (BE close to 530.5 eV [38,41,42], a weaker oxygen peak appears in the O 1s spectrum (at 529.9 eV), which is characteristic for the Bi–O bond [37].

XRD studies were conducted on all samples and the patterns are presented in Figure 3, Appendix A. Figure 3 shows a pattern of the Ta foil before (top) and after (bottom) the anodization process. Open circles represent experimental data points and the solid line is the LeBail model. As expected, only reflections of Ta metal are found for untreated foil. 

Ta reflections, but with different relative intensities, are also present after the anodization process. The estimated lattice parameter for Ta metal (Im-3m) a = 3.3258(3) Å is slightly larger than that reported in the literature [43]. There are only two strong XRD reflections present (marked by arrows) that cannot be attributed to the Ta phase. We have taken all 11 inorganic crystal sructure database (ICSD) records of Ta_2_O_5_ as a starting model for the LeBail analysis and the results have not been satisfied. This somehow surprising result might be explained by the low crystallinity or very thin layer of Ta_2_O_5_. The best fit was obtained for the Ta_4_O phase (Pmmm), with two reflections indexed by (300) and (400) indices, suggesting a strong preferential orientation. With two (h00) XRD peaks, only the a lattice parameter for Ta_4_O was refined, and the obtained value a = 7.220 Å is in good agreement with that reported in [38]. The relative intensity change of the XRD reflections of Ta metal might suggest that there are preferential crystallographic planes for the growth of a new phase. 

There are no evident changes in the XRD patterns as a result of using different voltages (Appendix A) and Ta_2_O_5_ NTs modified by Bi_2_S_3_ QDs (Appendix A). Pure Ta_2_O_5_ was found in a powder collected after calcination at 650 °C and 1000 °C (Appendix A).

The photocatalytic activity of NTs is presented in Appendix A. Based on the obtained results, it can be concluded that all of the samples presented highly efficient toluene photodegradation. To exclude the effect of toluene adsorption at the surface of the photocatalyst, the amount of adsorbed toluene was measured for the most active sample in the blank test (Appendix A). After 30 min in the dark, about 16% of toluene was adsorbed on the surface of Ta_2_O_5_ NTs. The samples were prepared under various circumstances, such as the applied voltage, cleaning the NTs after anodization in an ultrasonic bath, the presence of Bi_2_S_3_ QDs, the annealing atmosphere, the temperature, and the time of annealing, as well as the anodization procedure—one-step anodization vs. two-step anodization. Additionally, the results of the photodegradation of toluene after 5 min of irradiation are presented in Appendix A, where it can be seen that most of the toluene was decomposed within a short time. NTs obtained under a voltage in the range of 10–20 V indicated that the NTs obtained at 15 V showed the highest activity in the photodegradation of toluene. This may be affected by the NTs that possessed the optimal length, as well as strong adhesion. The nanotubes anodized at 20 V are less effective, despite their greater length. This may have been caused by detaching the nanotubes from the tantalum sheet during annealing or drying. The lowest photoactivity was observed for the sample anodized at 10 V, whose length was the shortest. Based on the above, it could be concluded that the optimal value of applied voltages during anodization of the Ta sheet is 15 V. Moreover, a comparison of the morphology of samples fabricated with and without cleaning in an ultrasonic bath indicated that the removal of initial oxide layers from the NT surface improved the photocatalytic activity of the material, because during cleaning by ultrasound, the compact oxide layer was removed from the NTs and the entire surface of the NTs was exposed for irradiation. Analysis of the dependence of annealing atmosphere and time on the photocatalytic properties of NTs indicated that the atmosphere during annealing displayed an insignificant role in the resultant photocatalytic properties of NTs. The photoactivity of NTs slightly decreases for annealing atmosphere in the following order: nitrogen, hydrogen, ammonia, and air. The differences in the photoactivity of samples annealed at various temperatures were very insignificant, i.e., 94.48% and 95.36% of toluene degradation efficiency for samples annealed at 300 °C and 450 °C, respectively. Moreover, increasing the temperature to over 450 °C causes the destruction of a sample with NTs (as discussed earlier), so the photoactivity of these samples was not measured. Based on the photoactivity results, it could be concluded that 1 h was the most optimal NTs annealing time, because samples annealed for 3 h were characterised by a lower photoactivity. The difference is not so significant; after 5 min of irradiation, the photodegradation of toluene was 94.20% and 94.08% for samples annealed at 300 °C and 450 °C, respectively. The least significant factor, which was analysed to find the most optimal condition for the fabrication of the NTs, was the synthesis of nanotubes by two-step anodization. This route of synthesis included removing the NT layer from the tantalum foil after anodic oxidation and subsequently performing the next anodization. Based on the results, it could be concluded that samples obtained using this one-step method were characterised by a higher photoactivity. The highest photoactivity of the obtained NTs is likely to be directly related to the length of NTs, equal to 1.15 and 1.27 nm for samples fabricated in one-step and two-step anodization, respectively. The efficiency of the toluene photodegradation in the range of 0–30 min is presented in Appendix A.

The modification of NTs by Bi_2_S_3_ QDs enhanced the photocatalytic properties of the material. It was observed that after 5 min of irradiation, almost 100% of toluene was degraded. Therefore, to prove that the modification of Ta_2_O_5_ by Bi_2_S_3_ QDs increased the photoactivity, additional experiments with a higher initial concentration of toluene (C_0_ = 400 ppm) were carried out. It was clearly observed that almost 100% of toluene was degraded after 30 min of irradiation over the Bi_2_S_3_ QD-modified Ta_2_O_5_ NTs, while the same duration of irradiation over pristine Ta_2_O_5_ NTs resulted in only 64% toluene degradation (Appendix A). However, the efficiency decreased with an increasing number of SILAR cycles. Based on the SEM images, it could be concluded that an increase in the number of SILAR cycles resulted in the aggregation of Bi_2_S_3_ particles on the NT surface, as shown in Figure 4. Consequently, the NT active layer was blocked by Bi_2_S_3_ particles and the irradiation was unable to penetrate the entire surface of the NTs. To confirm the presence of Bi_2_S_3_ QDs on the NT surface, TEM analysis was carried out. Figure 5 presents nanodots homogenously deposited on the wall of the NTs. The size of QDs is in the range of 10–11 nm and about 6 nm for 1–2 cycles and three SILAR cycles, respectively. It was not known that the size of nanoparticles decreased after three SILAR cycles. Typically, the size of nanoparticles increases when also increasing the number of SILAR cycles. Moreover, EDX analysis additionally confirmed the presence of Bi_2_S_3_ QDs (Appendix A). 

To evaluate the stability of the NTs during irradiation and their ability to regenerate after photocatalytic tests, the photodegradation of toluene during three cycles was investigated. The experiment was performed with and without regeneration of the photocatalyst by UV-Vis irradiation of the NTs for 1 h. Based on the results, it could be concluded that the photocatalytic activity of the unregenerated NTs decreased after each experiment (Figure 6a). This could have been caused by the surface being blocked by adsorbed by-products formed during toluene photodegradation [43,44]. However, the irradiation of the photocatalyst allowed NTs to regenerate and provided a constant efficiency of toluene degradation during irradiation. A similar dependence was observed for NTs modified by QDs. The results clearly indicated that NTs modified by QDs could be used several times for photocatalytic testing in the gas phase—however, the activities slightly decreased after each experiment. SEM analysis proved that the surface of the photocatalyst did not change, i.e., the nanotube structure was stable, after a few photocatalytic cycles (Appendix A).

The proposed mechanism of toluene photodegradation over Bi_2_S_3_ QDs/Ta_2_O_5_ NT composites is presented in Figure 7. Based on the literature [45], it could be concluded that the level of VB and CB of Ta_2_O_5_ is −3.9 and −7.8 eV in vacuum, respectively. Therefore, electrons of unmodified Ta_2_O_5_ NTs during UV-Vis irradiation were excited from the VB to its CB and the reactive pairs of electrons–holes were created. Then, electrons from CB were able to react with oxygen to create superoxide radicals O_2_^−^, because the potential required to produce O_2_^−^ is −4.17 eV in vacuum. Additionally, the position of the VB band of Ta_2_O_5_ is suitable for the creation of ^•^OH radicals from water present in the reaction system, because of humidity of the air. These reactive species were mainly responsible for the decomposition of toluene (Figure 7a). The band gap of Bi_2_S_3_ QDs is in the range of 1.3–1.7 eV. Moreover, the CB and VB level of the QDs is −4.1 and −5.7 eV in vacuum, respectively. Therefore, the mechanism of photodecomposition for NTs modified by Bi_2_S_3_ QDs (Figure 7b) is similar to the excitation of pristine nanotubes. The photoinduced holes of Ta_2_O_5_ reacted with water to create the OH. On the other hand, the photoexcited electrons from the CB of Ta_2_O_5_ reacted with oxygen, and then generated superoxide radicals O_2_^−^. Moreover, electrons from the CB of Ta_2_O_5_ migrated to the CB of Bi_2_S_3_ and finally, reacted with oxygen to produce superoxide radicals and decomposed toluene. Therefore, enhanced photoactivity of the Bi_2_S_3_ QDs/Ta_2_O_5_ NT composite could be attributed to the simultaneous excitation of both semiconductors and the separation of photoinduced pairs of electron–holes from Ta_2_O_5_ due to the migration of electrons from the CB of Ta_2_O_5_ to the CB of Bi_2_S_3_. To confirm the role of the Bi_2_S_3_ quantum dots in the photoexcitation step, experiments with monochromatic light were also performed. In those experiments, toluene was irradiated in the presence of Ta_2_O_5_ NTs and Bi_2_S_3_ QDs/Ta_2_O_5_ NTs using a wavelength from UV (318 nm) and a visible range (730 nm, which corresponds to the excitation of Bi_2_S_3_ QDs). It was observed that 71% of toluene was degraded after 30 min of irradiation by 318 nm over Ta_2_O_5_ NT, while samples modified by Bi_2_S_3_ QDs resulted in degradation of 89% of toluene. What is truly interesting is that we found that the irradiation of toluene over Bi_2_S_3_ QDs/Ta_2_O_5_ NTs using a 730 nm wavelength resulted in 39% toluene degradation. For unmodified samples, the toluene concentration decreased by about 14%, probably due to adsorption of the nanotube surface and photoreactor walls (Figure 6b). To exclude the effect of toluene photolysis (UV) and adsorption at the photoreactor walls (UV and Vis), additional reference experiments (without a photocatalyst) were performed. It was observed that 25% and 6% of toluene disappeared during irradiation by a wavelength equal to 318 and 730 nm, respectively. Therefore, the most remarkable result to emerge from this data is that the enhanced photoactivity of the Bi_2_S_3_ QDs/Ta_2_O_5_ NT composite results from the simultaneous excitation of Ta_2_O_5_ NTs and Bi_2_S_3_ QDs under UV and Vis irradiation, respectively. Moreover, the observed photoactivity could be correlated with the size of quantum dots. It is well-known that the band gap of quantum dots decreases upon an increase in their size [30]. This can explain the lower photoactivity of a sample containing smaller quantum dots, i.e., a narrower band gap increases the energy distance between semiconductors and, consequently, they cannot interact with each other effectively, but this point requires further investigation.

## 4. Conclusions

In summary, the adhered Ta_2_O_5_ nanotube arrays were obtained by the anodic oxidation of tantalum foil. The various factors, which could influence the morphology and stability of the photocatalyst, were analysed to find the most suitable condition for the synthesis of very photoactive layers for the decomposition of pollutants present in the gas phase. The most active samples were obtained under 15 V during the anodization of tantalum foil for 5 min. Subsequently, the NTs were cleaned in an ultrasonic bath and annealed in a nitrogen atmosphere for 1 h. This procedure allowed the nanostructures to be obtained, of which the external and internal diameters were 41 and 19 nm, respectively, and the thickness of the NT layer was 1.27 µm. Moreover, the photocatalyst could be utilized several times after regeneration. The modification of Ta_2_O_5_ NTs by Bi_2_S_3_ QDs enhances the photoactivity of the materials—however, when the amount of Bi_2_S_3_ in the sample is higher than 2.37 at. %, the photocatalytic properties decrease. 

## Figures and Tables

**Figure 1 nanomaterials-09-01347-f001:**
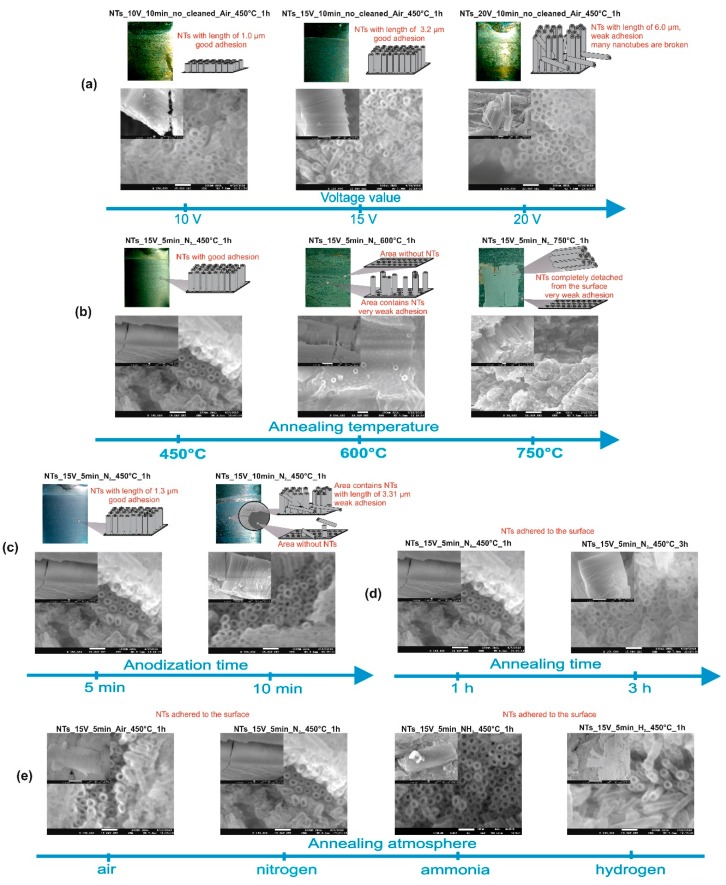
Scanning electron microscope (SEM) images of nanotube arrays (NTs): the effect of (**a**) applied voltage, (**b**) annealing temperature, (**c**) anodization time, and (**d**) annealing duration, (**e**) annealing atmosphere on the morphology and adhesive properties of tantalum pentaoxide (Ta_2_O_5_) nanotubes.

**Figure 2 nanomaterials-09-01347-f002:**
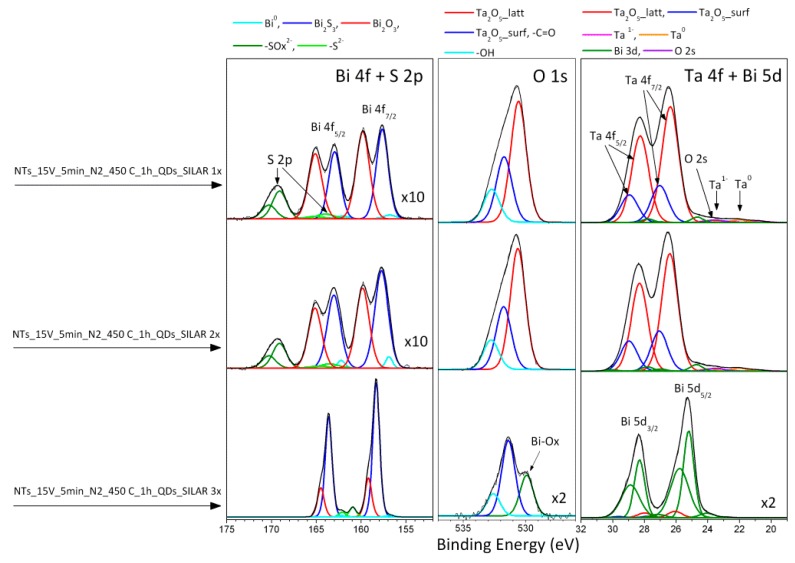
X-ray photoelectron spectroscopy (XPS) spectra of nanotube array (NT)-modified bismuth sulfide (Bi_2_S_3_) quantum dots (QDs).

**Figure 3 nanomaterials-09-01347-f003:**
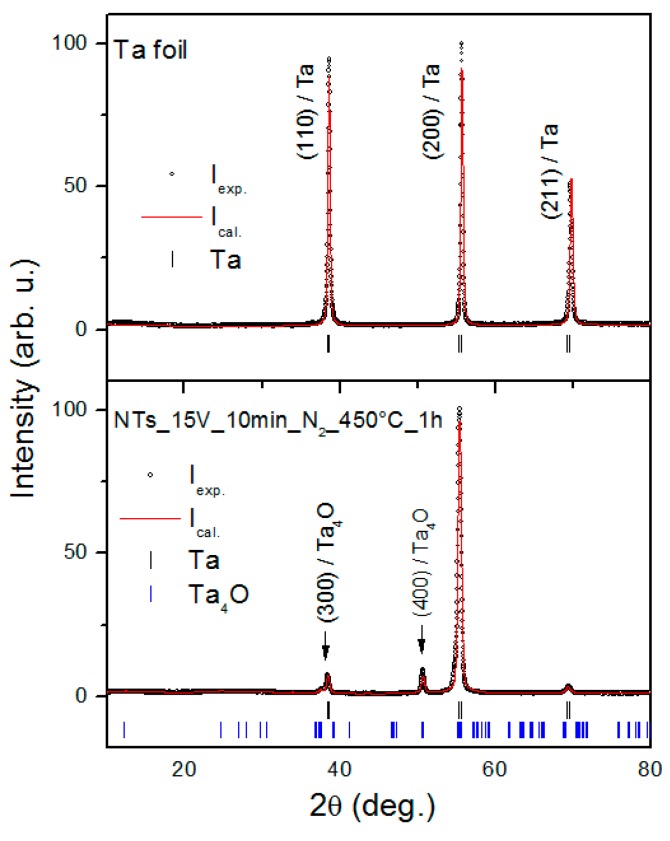
X-ray diffraction patterns for the Management Information Base (MIB) sample before (top) and after (bottom) the anodization process. Open circles represent experimental data, whereas the solid red line is a LeBail profile fit with two models used: Ta (Im-3m) and Ta_4_O (Pmmm) shown by black and blue vertical bars, respectively.

**Figure 4 nanomaterials-09-01347-f004:**
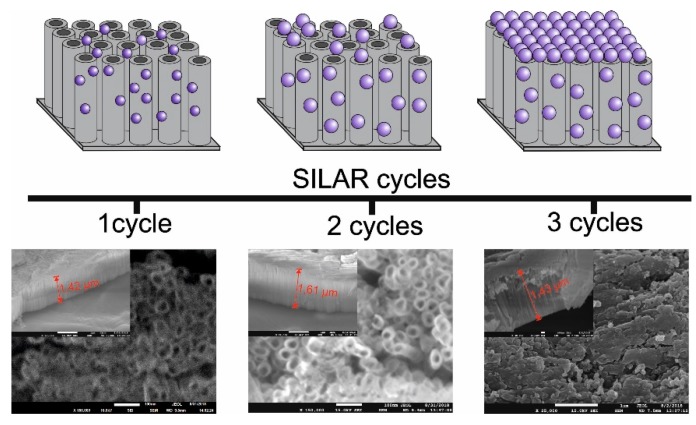
Scanning electron microscope (SEM) images and schematic representation of the surface of nanotube arrays (NTs) modified by quantum dots (QDs).

**Figure 5 nanomaterials-09-01347-f005:**
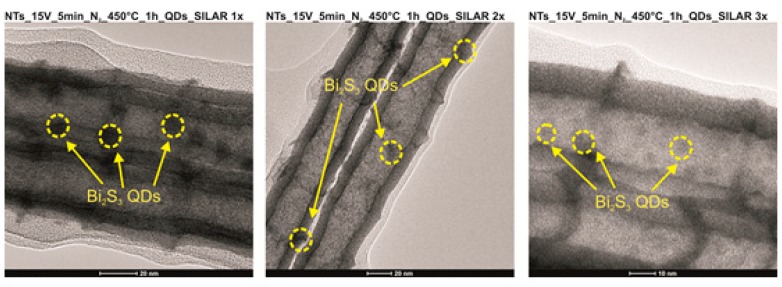
Transmission Electron Microscopy (TEM) images for nanotube arrays (NTs) modified by bismuth sulfide (Bi_2_S_3_) quantum dots (QDs).

**Figure 6 nanomaterials-09-01347-f006:**
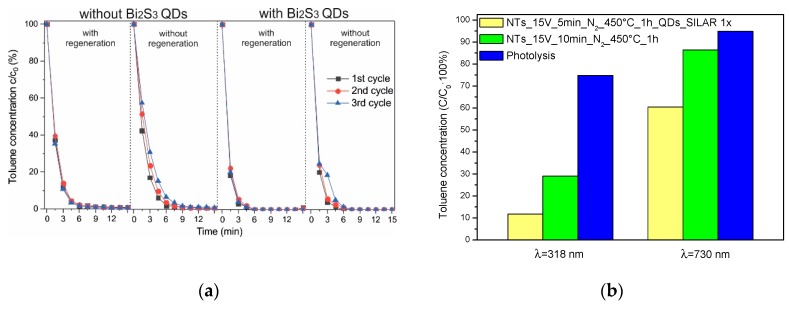
Photocatalytic activity of selected samples, shown as toluene degradation: (**a**) photocatalyst stability during three irradiation cycles for NTs_15 V_5 min_N_2__450 °C_1 h and NTs_15 V_5 min_N_2__450 °C_1 h_QDs_SILAR 1x. (**b**) Efficiency of toluene removal over Ta_2_O_5_ nanotube arrays (NTs) and bismuth sulfide (Bi_2_S_3_) quantum dots (QDs)/Tantalum pentoxide (Ta_2_O_5_) NTs samples and in the blank tests (toluene photolysis in the absence of a photocatalyst) under monochromatic light—318 nm and 730 nm.

**Figure 7 nanomaterials-09-01347-f007:**
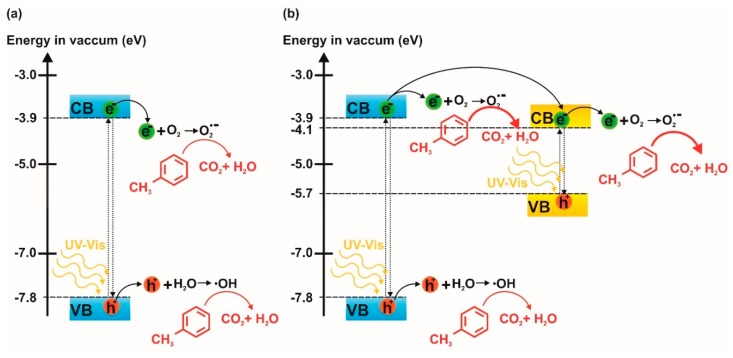
Proposed photodegradation mechanism for (**a**) nanotube arrays (NTs) and (**b**) NTs modified by quantum dots (QDs).

**Table 1 nanomaterials-09-01347-t001:** Sample labels, synthesis conditions, morphology, and photoactivity of bare and quantum dot (QD)-decorated nanotube arrays (NTs).

Sample Label	Preparation Conditions During Anodic Oxidation (AO)	External Diameter (nm)	Internal Diameter (nm)	Thickness (nm)	Length (µm)	Ta_2_O_5_ NTs Adhesion to Ta Foil	Toluene Decomposition (%)	
							5 min of Irradiation	15 min of Irradiation
NTs_10 V_10 min_no_cleaned_Air_450 °C_1 h	AO (U = 10 V, t = 10 min.), dried (T = 80 °C, t = 24 h), and annealed in air (T = 450 °C; t = 1 h)	44 ± 3	27 ± 2	9 ± 1	1.00 ± 0.13	High	64.22 ± 3.27	90.51 ± 3.27
NTs_15 V_10 min_ no_cleaned Air_450 °C_1 h	AO (U = 15 V, t = 10 min.), dried (T = 80 °C, t = 24 h), and annealed in air (T = 450 °C; t = 1 h)	46 ± 6	24 ± 2	11 ± 2	3.18 ± 0.09	High	93.55 ± 2.38	97.83 ± 1.29
NTs_20 V_10 min_ no_cleaned Air_450 °C_1 h	AO (U = 20 V, t = 10 min.), dried (T = 80 °C, t = 24 h), and annealed in air (T = 450 °C; t = 1 h)	48 ± 3	28 ± 3	10 ± 1	6.00 ± 0.19	Weak	91.05 ± 3.35	96.82 ± 2.45
NTs_10 V_10 min_N_2__450 °C_1 h	AO (U = 10 V, t = 10 min.), ultrasonically cleaned (1 min.), dried (T = 80 °C, t = 24 h), and annealed in N_2_ (T = 450 °C; t = 1 h)	45 ± 5	20 ± 3	11 ± 2	1.74 ± 0.02	Weak	68.38 ± 0.35	94.65 ± 1.91
NTs_15 V_5 min_N_2__450 °C_1 h	AO (U = 15 V, t = 5 min.), ultrasonically cleaned (1 min.), dried (T = 80 °C, t = 24 h), and annealed in N_2_ (T = 450 °C; t = 1 h)	41 ± 4	19 ± 3	10 ± 2	1.27 ± 0.05	High	95.36 ± 1.22	98.73 ± 0.47
NTs_15 V_5 min_N_2__450 °C_3 h	AO (U = 15 V, t = 5 min.), ultrasonically cleaned (1 min.), dried (T = 80 °C, t = 24 h), and annealed in N_2_ (T = 450 °C; t = 3 h)	48 ± 2	25 ± 4	10 ± 1	1.78 ± 0.11	High	91.03 ± 0.47	98.98 ± 0.06
NTs_15 V_5 min_N_2__600 °C_1 h	AO (U = 15 V, t = 5 min.), ultrasonically cleaned (1 min.), dried (T = 80 °C, t = 24 h), and annealed in N_2_ (T = 600 °C; t = 1 h)	39 ± 2	23 ± 3	10 ± 2	1.21 ± 0.03	Very weak	Sample was unstable	
NTs_15 V_5 min_N_2__750 °C_1 h	AO (U = 15 V, t = 5 min.), ultrasonically cleaned (1 min.), dried (T = 80 °C, t = 24 h), and annealed in N_2_ (T = 750 °C; t = 1 h)	40 ± 3	21 ± 2	10 ± 1	1.43 ± 0.02	Very weak	Sample was unstable	
NTs_15 V_10 min_N_2__450 °C_1 h	AO (U = 15 V, t = 5 min.), ultrasonically cleaned (1 min.), dried (T = 80 °C, t = 24 h), and annealed in N_2_ (T = 450 °C; t = 1 h)	42 ± 3	21 ± 2	10 ± 2	3.31 ± 0.08	High	92.74 ± 1.09	96.81 ± 0.35
NTs_15 V_5 min_Air_450 °C_1 h	AO (U = 15 V, t = 5 min.), ultrasonically cleaned (1 min.), dried (T = 80 °C, t = 24 h), and annealed in air (T = 450 °C; t = 1 h)	49 ± 7	23 ± 4	10 ± 2	1.46 ± 0.20	High	91.86 ± 2.16	97.50 ± 2.40
NTs_15 V_5 min_N H_3__450 °C_1 h	AO (U = 15 V, t = 5 min.), ultrasonically cleaned (1 min.), dried (T = 80 °C, t = 24 h), and annealed in NH_3_ (T = 450 °C; t = 1 h)	47 ± 5	19 ± 3	13 ± 1	2.19 ± 0.07	High	93.37 ± 0.42	98.90 ± 0.00
NTs_15 V_5 min_ H_2__450 °C_1 h	AO (U = 15 V, t = 5 min.), ultrasonically cleaned (1 min.), dried (T = 80 °C, t = 24 h), and annealed in H_2_ (T = 450 °C; t = 1 h)	44 ± 6	21 ± 3	10 ± 1	1.68 ± 0.03	High	94.20 ± 3.43	98.09 ± 0.03
NTs_15 V_5 min_N_2__300 °C_1 h	AO (U = 15 V, t = 5 min.), ultrasonically cleaned (1 min.), dried (T = 80 °C, t = 24 h), and annealed in N_2_ (T = 300 °C; t = 1 h)	47 ± 5	24 ± 3	11 ± 1	2.25 ± 0.06	High	94.08 ± 0.42	97.36 ± 0.25
NTs_15 V_5 min_N_2__450 °C_1 h_two_step	AO (I step: U = 15 V, t = 5 min.), removing of NTs layer, AO (II step, U = 15 V, t = 5 min.), ultrasonically cleaned (1 min.), dried (T = 80 °C, t = 24 h), and annealed in N_2_ (T = 450 °C; t = 1 h)	46 ± 5	24 ± 4	10 ± 1	1.15 ± 0.05	Weak	89.28 ± 0.14	96.53 ± 0.30
NTs_15 V_5 min_N_2__450 °C_1 h_QDs_SILAR 1x	AO (U = 15 V, t = 5 min.), ultrasonically cleaned (1 min.), dried (T = 80 °C, t = 24 h), and annealed in N_2_ (T = 450 °C; t = 1 h), 1 cycle of SILAR	50 ± 4	31 ± 3	10 ± 1	1.42 ± 0.03	High	99.17 ± 0.14	100
NTs_15 V_5 min_N_2__450 °C_1 h_QDs_SILAR 2x	AO (U = 15 V, t = 5 min.), ultrasonically cleaned (1 min.), dried (T = 80 °C, t = 24 h), and annealed in N_2_ (T = 450 °C; t = 1 h), 2 cycles of SILAR	41 ± 4	28 ± 3	8 ± 1	1.61 ± 0.11	High	71.56 ± 1.12	100
NTs_15 V_5 min_N_2__450 °C_1 h_QDs_SILAR 3x	AO (U = 15 V, t = 5 min.), ultrasonically cleaned (1 min.), dried (T = 80 °C, t = 24 h), and annealed in N_2_ (T = 450 °C; t = 1 h), 3 cycles of SILAR	Coated with Bi_2_S_3_ layer			1.40 ± 0.05	High	Inactive	

**Table 2 nanomaterials-09-01347-t002:** Elemental composition (in at. %) of the surface layer of NTs_15V_5min_N_2__450 °C samples before and after subsequent SILAR cycles. The chemical states of Bi and Ta (in %) were evaluated after deconvolution of the Bi 4f and Ta 4f X-ray photoelectron spectroscopy (XPS) spectra, respectively.

Sample Label	Elemental Composition (at. %)						Bi 4f_7/2_ Fractions (%)			Ta 4f_7/2_ Fractions (%)			
	Ta	O	C	S	Bi	Residue (F, N, Na)	Bi^0^ 157.0 ± 0.2 eV	Bi_2_S_3_ 158.0 ± 0.3 eV	Bi_2_O_3_ 159.5 ± 0.3 eV	Ta_2_O_5__Surf 27.0 ± 0.1 eV	Ta_2_O_5_ 26.2 ± 0.2 eV	Ta^1+^ 22.1 ± 0.1 eV	Ta^0^ 21.0 ± 0.3 eV
NTs_15 V_5 min_N_2__450 °C_1 h	22.44	46.00	24.31	3.10	-	4.15	0	0	0	0	96.84	1.72	1.44
NTs_15 V_5 min_N_2__450 °C_1 h_QDs_SILAR 1x	15.79	58.20	11.02	6.26	2.37	6.36	2.28	46.50	51.22	23.55	73.42	1.86	1.17
NTs_15 V_5 min_N_2__450 °C_1 h_QDs_SILAR 2x	16.24	57.47	12.13	5.66	2.63	5.87	3.21	53.10	43.69	24.81	72.51	1.91	0.77
NTs_15 V_5 min_N_2__450 °C_1 h_QDs_SILAR 3x	0.63	29.47	26.72	15.14	22.60	5.44	1.24	76.58	22.18	19.61	80.39	0	0

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
