# Peer review of "Fabrication of Durable Ordered Ta2O5 Nanotube Arrays Decorated with Bi2S3 Quantum Dots"

_nanomaterials, 2019, doi:10.3390/nano9101347_

Round 1

Reviewer 1 Report

This manuscript reported the fabrication of durable ordered Ta2O5 nanotube arrays decorated by Bi2S3 quantum dots. However, the photdegradation results are not significant, and the provided data are insufficient. Therefore, I cannot recommend this manuscript to be published.

Comments:

1. The structural evolution of anodic Ta2O5 nanotubes (such as XRD data) with different applied voltages, anodization times, and annealing temperatures should be provided.

2. The XRD data of Bi2S3 nanoparticles with three cycles of SILAR process should be supplied and discussed.

3. The authors should explain why the nanoparticle size of 3 cycles SILAR is smaller that of 2 and 1 cycle SILAR. Typically, the nanoparticles increase the size with SILAR cycle number.

4. The authors should discuss the limited nanoparticles size for the quantum size effect of Bi2S3. In addition, the evidences for quantum size effect of Bi2S3 should be provided.

5. Actually, the photodegradation results with and without Bi2S3 are not significant. 

Reviewer 2 Report

The manuscript by Baluk et al. report a fabrication of Ta2O3 nanotube photocatalysts and their photocatalytic activities on gas-phase toluene degradation. Authors varied several synthetic parameters and synthesized various Ta2O3 nanotube materials. Although authors tried to discuss their results and to explain their clams, it is hard to find a deep discussion such as relationship between synthetic parameters, property change and photocatalytic activity. They just mentioned which is better or the best. I don’t think that the manuscript is ready for publication in current version. The authors should need to update the manuscript with following major revisions.

(1) I am wondering why authors introduced Bi2S3 QD materials. Although Bi2S3 QD-Ta2O3 NT showed the slightly higher photocatalytic activity than pure Ta2O3 NT, the difference was small. (It seems be a negligible difference, 95.36% for pure NT and 99.17% for Bi2S3/Ta2O3 NT. It is less than 4% difference for 5 min). Since authors used 1000 W UV-vis light and electron exiting could mainly be happened in Ta2O3 NT, the difference of photocatalytic activity over the above two catalysts should be small. If authors use only visible light irradiation using 420 cut-off filter, they can find the advantage of Bi2S3 QDs-Ta2O3 NT compared to pure Ta2O3 NT. I recommend that authors should do photocatalytic activity test under visible light conditions (> 420 nm) and make discussion about performance enhancement of Bi2S3/Ta2O3 NT.

(2) In addition, authors can make additional paragraph about Bi2S3 QD materials in the field of photocatalysis and photocatalytic performance in introduction section.

(3) In XRD patterns, authors displayed one XRD pattern of Ta2O3 NT/Ta plate. It seems that authors just explain peak assignments based on LeBail Model. I recommend that authors can update the XRD pattern of pure Ta plate. Maybe authors can find obvious differences between Ta2O3 NT/Ta plate and pure Ta plate and make additional discussion.

(4) Authors claimed stability of Ta2O3 NT in the manuscript. To claim structural stability of Ta2O3 NT after reaction, I recommend that they can update the SEM data after several cycles of photocatalysis reaction.

(5) Even though authors claimed that NTs_15V_5min_N2_450°C_1h showed the best performance among the NTs employed, it is difficult to find the reason of high performance and relationships between catalyst property and high performance. Which characteristic is the important factor in photocatalytic activity?

(6) In supporting information, I recommend that authors should update XPS spectra data of 4 kind of pure NT samples and make more discussion.

Reviewer 3 Report

The present ms from KobylaÅ„ski, Zaleska-Medynska and co-workers describes the synthesis of Ta2O5 nanotubes, their decoration with Bi2S3, and photocatalytic testing for toluene degradation. I find the contribution interesting, albeit it has to be strongly improved before final publication. In particular, attention to the issues indicated below should be paid.

- The style is verbose and I strongly suggest the shorten the ms significantly (20-30%); on top of that, the language is rather poor and extensive revision by a native speaker is required;

- As for the photocatalytic tests, I recommend the Authors to better define the conditions of the experiments; in particular, what happens during the period in the dark (30 min. to achieve adsorption/desorption equilibrium)? Is there any variation of toluene concentration? How the Authors can assume that toluene is degraded and not simply adsorbed?

- As for the mechanism, it seems that the two materials (Ta2O5 NTs and Bi2S3) does not interact with each other, as apparent from the position of the bands. Is this the case? No electron/hole transfer between the two is possible? If this is so, which is the rationale of combining them together? To address this point, I recommend to perform experiments with monochromatic light sources (emitting only in the UV or only in the VIS range), in order to appreciate the contribution of Ta2O5 alone and of Bi2S3 alone.

- Minor point: please define all the acronyms at their first appearance.

Round 2

Reviewer 1 Report

The authors have adequately answered to my main remarks. My opinion is that the resubmitted paper could be considered for publishing on Nanomaterials without any further revision.